# Fire Suppression and Thermal Behavior of Biobased Rigid Polyurethane Foam Filled with Biomass Incineration Waste Ash

**DOI:** 10.3390/polym12030683

**Published:** 2020-03-19

**Authors:** Agnė Kairytė, Arūnas Kremensas, Saulius Vaitkus, Sylwia Członka, Anna Strąkowska

**Affiliations:** 1Faculty of Civil Engineering, Institute of Building Materials, Laboratory of Thermal Insulating Materials and Acoustics, Vilnius Gediminas Technical University, Linkmenu st. 28, LT-08217 Vilnius, Lithuania; arunas.kremensas@vgtu.lt (A.K.); saulius.vaitkus@vgtu.lt (S.V.); 2Institute of Polymer and Dye Technology, Lodz University of Technology, Stefana Żeromskiego 116, 90-924 Lodz, Poland; sylwia.czlonka@edu.p.lodz.pl (S.C.); anna.strakowska@p.lodz.pl (A.S.)

**Keywords:** polyurethane foam, fire resistance, thermal stability, carbon footprint, biomass waste ash

## Abstract

Currently, there is great demand to implement circular economy principles and motivate producers of building materials to integrate into a closed loop supply chain system and improve sustainability of their end-product. Therefore, it is of great interest to replace conventional raw materials with inorganic or organic waste-based and filler-type additives to promote sustainability and the close loop chain. This article investigates the possibility of bottom waste incineration ash (WA) particles to be used as a flame retardant replacement to increase fire safety and thermal stability under higher temperatures. From 10 wt.% to 50 wt.% WA particles do not significantly deteriorate performance characteristics, such as compressive strength, thermal conductivity, and water absorption after 28 days of immersion, and at 32 °C WA particles improve the thermal stability of resultant PU foams. Furthermore, 50 wt.% WA particles reduce average heat release by 69% and CO_2_ and CO yields during fire by 76% and 77%, respectively. Unfortunately, WA particles do not act as a smoke suppressant and do not reduce smoke release rate.

## 1. Introduction

Nowadays, it is accepted that biofuel incineration does not promote the greenhouse effect because of the neutral carbon dioxide conversion, which is determined by the lignocellulosic biomass renewability. The attention to bioenergy as an alternative to fossil fuel-based energy has significantly increased in recent years due to questions regarding the global warming mostly originating from fossil fuel combustion. It was distinguished that the utilization of biomass resources will be one of the most important factors for environmental protection in the 21st century [1,2]. The growing use of biomass-based fuels for energy generation will lead to large amounts of biomass incineration ash, which causes serious environmental problems. Currently, a major part of this waste obtained in Lithuania is disposed in landfills. As biomass is highly significant in a circular economy scenario, it requires that no unrecovered waste occurs. Therefore, it is of great demand to find alternative applications of waste ash (WA).

Some of the major studies are based on production of structural materials and WA utilization as a replacement for major raw materials. An overview of recent studies [3] showed that WA is acceptable as an alternative replacement to cements and concrete aggregates due to its high silica content [4,5]. It was shown that WA could be used as a pozzolan up to 20 wt.% [6,7] and as an additive up to 10 wt.% [8] to obtain concrete material with optimal characteristics as well as to reduce price and CO_2_ emissions [9]. However, it is determined [10] that such WA has averagely only 6.30 wt.% of spherical particles; therefore, its usage as a pozzolanic additive in concretes is limited.

Another type of building material where WA could be utilized and is starting to be researched is polyurethane (PU) foam. It is lightweight and has extremely valuable performance characteristics: low thermal conductivity, water absorption, and sufficient strength. However, the flammability of PU foam limits its further application in areas with increasing fire resistance requirements. Therefore, over the years, researches concerning the reduction of PU flammability have attracted a lot of attention at the national and international level.

For a long time, halogenated compounds were very important components of PU foams as flame retardants because of their effectiveness and low price. Unfortunately, when burnt halogenated compounds emit toxic smoke, thus polluting the environment and damaging human health. As environmental awareness develops, many halogen-free flame retardants have been used to improve the fire resistance of PU foams and their composites, e.g., phosphorus compounds [11,12,13], intumescent flame retardants [14,15], metal hydroxide [16,17], and waste-based and filler-type additives [18,19]. Some of the waste-based and filler-type additives consist of inorganic particles that can act as a flame spread barrier and improve fire resistance characteristics, such as heat release rate, as well as shift towards higher thermal degradation temperature [20]. However, such inorganic waste-based and filler type additives can be used in rather low amounts, because they increase viscosity, cause cell rupture, and deteriorate performance characteristics [21,22]. Researchers [23] observed that the addition of up to 40 wt.% of basalt waste reduces the compressive strength of PU composites and worsens their dimensional stability, whereas others [24] tested feldspar and kaolinite clay and showed that even 10 wt.% reduces mechanical performance of the resultant PU foams. However, the authors of [10] showed that WA particles as waste-based and filler-type additive can be used in higher amounts, i.e., up to 50 wt.%, because they have a lower impact on the polyol premix compared to other filler-type additives.

Therefore, the aim of present research is to modify PU foam with 10–50 wt.% WA particles and determine the impact of high amounts of WA particles on the fire resistance and thermal behavior of neat and WA particles-based PU foams, taking into consideration the microstructural analysis and a study of basic performance characteristics as well.

## 2. Materials and Methods

### 2.1. Materials

As the main components in the production of PU and WA particles modified PU (WAPU) foams, rapeseed oil-based polyol BioPolyol RD (SIA PolyLabs, Riga, Latvia) and Petol PZ 400-4G (Oltchim, Râmnicu Vâlcea, Romania) were used. They have hydroxyl values of 350 mg KOH/g and 421 mg KOH/g and water contents of <0.2% and <0.1%, respectively. A polymeric 4,4′-diphenylmethane diisocyanate Lupranat M20S (BASF, Ludwigshafen, Germany) was used as the second main component in the synthesis of PU and WAPU foams. Distilled water was used as a blowing agent. To catalyze reactions, blowing and gelling catalyst Polycat 9 (Air Products and Chemicals, Inc., Allentown, PA, USA) was added. For stabilization of foams, silicone surfactant ST-52 (Shijiazhuang Chuanghong Technology Co., Ltd., Shijiazhuang, China) was implemented. The WA was gained from the Lithuanian wood industry (JSC “Giriu bizonas”, Kazlu Ruda, Lithuania) from the grate and primary combustion chambers and used as the flame retardant filler without further modification or preparation [10]. The elemental composition of WA particles is as follows; C (5.80%), O (49.2%), Na (2.47%), Mg (2.05%), Al (3.59%), Si (15.7%), K (3.67%), Ca (10.3%), and Fe (7.22%).

### 2.2. Preparation of PU and WAPU Foams

The selected amounts of components for the synthesis of PU and WAPU foams are presented in Table 1. PU and WAPU foams were produced by the one-step method from a two-component system, with an equivalent ratio of NCO:OH groups equal to 1.25:1.

For the preparation of dimensionally stable PU and WAPU foams, a system consisting of 60% of rapeseed-based and 40% of sucrose-based biopolyols was used. The mixture of these two biopolyols was then modified with scaled amounts of water, catalyst, and surfactant and thoroughly mixed at 1800 rpm speed for 1 min. Further, 10 wt.%, 30 wt.%, and 50 wt.% of WA were added into the prepared mixture. Then, a weighted amount of diisocyanate was poured into a viscous WA-modified mixture and mixed for another 10 s. Next, the mixture was poured into (300 × 300 × 100) mm^3^ forms and left to foam at 23 ± 2 °C temperature. The prepared specimens were kept for 24 h before being demolded and cut. Before the tests, specimens were maintained at a standardized conditioning temperature of 23 ± 5 °C and relative air humidity of 50 ± 5% for 24 h to settle.

### 2.3. Test Methods

Specimens of 50 × 50 × 50 mm^3^ in size were used to determine the linear dimensions of PU and WAPU foams according to work in [25] and the apparent density according to work in [26]. Thermal conductivity tests were carried out for 300 × 300 × 50 mm^3^ specimens based on the requirements of [27]. The test was carried with FOX 304 (“LaserComp”, Saugus, MA, USA). Its measurement limits vary from 0.01 W/(mK) to 0.50 W/(mK) and a measuring accuracy is approximately 1%. The difference between measuring plates was 20 °C and the average tests temperature 10 °C. The percentage of closed cells was determined according to method 2 in [28] for specimens of 100 × 30 × 30 mm^3^ in size. Water absorption by total immersion for 28 days was determined based on the [29] 2A method requirements for specimens of 200 × 200 × 50 mm^3^ in size. The compressive strength test was processed according to work in [30] for specimens of 50 × 50 × 50 mm^3^ in size with a testing machine H10KS (“Hounsfield”, Surrey, UK).

For the evaluation of the impact of WA particles on PU and WAPU foams under high temperatures, thermogravimetry (TGA) and differential thermogravimetry (DTG) were conducted under the air atmosphere using STA 449 F1 Jupiter Analyzer (“Netzsch Group”, Erlangen, Germany) at temperature interval from 25 to 600 °C. Temperature raising speed was 10 °C/min.

The limited oxygen index (LOI) was obtained using an Oxygen Index Instrument (“NETZSCH TAURUS Co.”, Ltd., Weimar, Germany). The size of the samples was (120 × 10 × 10) mm^3^ (length × width × thickness). A sample tip was ignited for 5 s by means of a gas burner supplied with a propane–butane mixture. The limited oxygen index was calculated as the percentage of oxygen and nitrogen volume in the mixture.

The reaction to fire of PU and WAPU was determined in accordance with [31] using a Cone 2a (“Atlas Electric Devices Co.”, Chicago, IL, USA) cone calorimeter. The main parameters during the test are heat flow—25 kW/m^2^, the area of tested surface—88.4 cm^2^, and testing time—376 s. Individual results of average heat emission (AVHR), total smoke released (TSR), yields of carbon monoxide (COY), and carbon dioxide (CO_2_Y) were obtained and average values were calculated.

Energy dispersion spectroscopy (EDS) results were determined using scanning electron microscope (SEM) QUANTA 250 (“FEI Company”, Hillsboro, OR, USA) with EDS analyzer and INCA analytical system. Local chemical composition was determined by electron microscope electron beam excited characteristic X-ray spectra.

The structure of PU and WAPU after cone calorimetry test was obtained with SEM Helios NanoLab 650 (“FEI Company”, Hillsboro, OR, USA) with resolution of 0.8 nm and magnification varying from 50 to 1,000,000 times. During the test, a 3–4 kV voltage was used. The microstructure and its parameters were identified from the surface, which is parallel to the foaming direction. Before the SEM imaging, specimens were coated with a thin golden layer under a vacuum environment.

## 3. Results and Discussion

### 3.1. The Main Physical and Mechanical Properties of PU and WAPU Foams

It is well known that product standards, in this case the works in [32] and [33], define the main performance characteristics, which have to be determined and assured for PU foams. Such characteristics include mechanical properties, thermal, and water resistance. Therefore, Table 2 presents the main performance characteristics determined for PU and 10 wt.%, 30 wt.%, and 50 wt.% of WA-modified foams.

The apparent density of polymeric foam depends on the amount of the base raw material used to form a network, the density of the material that constitutes the foam matrix, and the density of the gas in cells [34]. In this case, WA particles purposefully interact with the WAPU foam matrix (Figure 1) and affect apparent density in their own way, i.e., 10 wt.% of WA in WAPU-10 foam averagely increase apparent density by 22% compared to that of a neat PU, whereas 30 wt.% and 50 wt.% of WA increase apparent density by 34% and 49%, respectively. According to the work in [21,22,35,36,37], organic and inorganic fillers increase dynamic viscosity of liquid mixtures which hinders the blowing efficiency of the foams.

It can be noticed that WA particles do not significantly increase the thermal conductivity of WAPU-50 foam and it is only a 7% higher increase than that of the PU foam. Even though the cell diameter of the foams decreases with the addition of WA particles, thermal conductivity increases. As the authors of [38] stated, this phenomenon can be due enhanced effect of thermal conductivity through the solid parts, in this case WA particles and thicker cell walls, as the cell size decreases under the same blowing agent contents. However, contrary results are observed in [18] when the thermal conductivity is tested 1 day after production. It is noticed that PU foams are characterized by aging, and therefore CO_2_ molecules are replaced over time by air molecules. Consequently, the thermal conductivity of PU and WAPU foams reach their limiting values.

Compressive strength is a crucial parameter for PU foams, and therefore we present the variation in parameter values under the different amounts of WA particles in Table 2. It is noted that at 10 wt.% and 30 wt.% of WA particles compressive strength compared to PU foam increased by 8.3% and 4.2%, respectively. The same amounts of corn–stover lignin were used to obtain similar variations in compressive strength [39]; however, the authors of [40] determined that the addition of steel slag even at small amounts significantly reduced the mentioned parameter and, vice versa, results are obtained by the authors of [41] for foams with carbon black showing that the nature, size, and shape of an added filler has a great influence on the final properties of PU foams.

As with almost all fillers [42,43,44], water absorption in polyurethane foams increases. The same results were obtained in present study. It can be noticed that water absorption at 10 wt.%, 30 wt.%, and 50 wt.% of WA particles increases by 5.8%, 11%, and 25%, respectively.

In all cases, WA particles act as a defect-provoking material, causing cracks, a poor interface between the particles and polymer matrix, as well as forming voids and interconnected and open pores, which allow water penetration.

### 3.2. Thermal Stability of PU and WAPU Foams

The thermal degradation of PU foams is a heterogeneous process consisting of multiple physical and chemical reactions. This process attracts scientific and engineer communities because it gives an opportunity to obtain a fingerprint of rational or even optimal conditions to design and produce high quality end use PU foams.

TGA and DTG results are presented in Figure 2, and the main weight losses as well as temperature peaks can be observed in Table 3. Mass loss up to 150 °C is associated with moisture evaporation. The degradation temperature at 5 wt.% weight loss (T_5 wt.%_) is 213 °C, which is accompanied with three-stage degradation peaks. The first degradation interval ranges from 155 °C to 227 °C with a peak of 211 °C being assigned to degradation of the hard segments, the second interval ranges from 227 to 438 °C, and third interval ranges from 438 to 498 °C with peaks at 309 and 462 °C, respectively, which are assumed to be due to degradation of soft segments [45]. Additionally, 28.2 wt.% of char at 600 °C is obtained.

The same as PU foam, WAPU foams are characterized by three-stage degradation; however, the third degradation occurs at a different temperature range and is marked as the fourth stage in Table 3. It can be observed that at 10 wt.% of WA particles T_5 wt.%_ is 32 °C higher than the neat PU foam; the same tendency can be noted for all WA-modified WAPU and fly ash-modified PU foams in other authors work [45]. This difference means that WA particles are capable of shifting T_5 wt.%_ degradation temperature towards higher temperature, reduce weight loss, and increase thermal stability of WAPU foams. This behavior can be attributed to the barrier effect provided by some filler particles, including WA [46,47,48]. Additionally, a slight positive effect can be observed in the first stage. Thermal degradation interval of WAPU-10 foam is in the range of 174 to 239 °C with a peak at 211°C while WAPU-30 and WAPU-50 have the first stage degradation interval of (147 ÷ 246) °C with a peak at 222 and 159 to 241 °C with a peak at 217 °C, respectively. It is seen that 10 wt.% of WA particles shifted the stage one thermal degradation interval towards higher temperature by delaying the escape of volatile degradation products from PU [48].

The same decomposition profile can be observed for second stage intervals. WA particles have resulted into WAPU foams with the thermal degradation interval 239 to 430 °C with a peak temperature at 309 °C for WAPU-10 foam, 246 to 430 °C with a peak temperature of 308 °C for WAPU-30 and 241 to 421 °C with a peak temperature at 217 °C for WAPU-50. Therefore, WAPU foams show better stability of soft segments in the second degradation stage compared to that of PU foam. Note that the degradation rate of WAPU in Figure 2b is much lower compared to PU foam, i.e., changes from 0.0020%/°C for PU foam to 0.0012%/°C for WAPU-50 (first stage), and from 0.0035%/°C for PU foam to 0.0025%/°C for WAPU-50 (second stage). This effect is associated with to the assumption that fillers absorb part of the heat generated [47].

Moreover, char yield at 600 °C is greater for WAPU foams than PU foams. A significant difference of more than 12 wt.% can be observed for WAPU-50 foam compared to PU foam. These phenomena can be explained by the composition of WA particles, which shows that they consist mainly of compounds that degrade at a temperature higher than 600 °C such as CaCO_3_, Al_2_O_3_, SiO_2_, etc. The fourth stage can be also dedicated to the composition of WA particles. According to previous study [10], the content of organic matter is 4 wt.%; therefore, the peaks at ~585 °C in WAPU foams are associated to the degradation of woody part in WA particles.

### 3.3. Flame Retardancy of PU and WAPU Foams

The cone calorimetry testing can simulate a small scale fire and emulate a real scale fire. The test method was further used to determine the flammability of PU and WAPU foams. The relevant parameters and their numerical values are presented in Figure 3 and Table 4.

The AVHR curve during the cone calorimetry test of the PU foam exhibits a broader shoulder with a peak value of 136 kW/m^2^ (Table 4), compared to WAPU foams. In the case of the WAPU foams, the amount of 10 wt.% of WA particles does not considerably impact heat emission. However, twice better performance can be observed for WAPU foams with 30 wt.% and 50 wt.%, i.e., WA particles reduced AVHR value by approximately 44% and 69%, respectively. Such an effect could be assigned to the development of a char residue layer [49,50]. The formation of a char layer on the surface protects it against heat penetration and release of combustible materials. Thus, the combustion was effectively suppressed by the addition of WA particles. Similar observations were done by the authors of [51] who studied polyurethane foam fire retarded with melamine pyrophosphate. It is worth mentioning that the improvement in AVHR values is connected to the presence of inflammable components such as CaCO_3_. This chemical compound undergoes thermal decomposition between 600 and 800 °C [52,53]; it transforms into calcium oxide and releases CO_2_. The originating endothermic reaction induces local cooling and the O_2_ in the flame zone is diluted by the released CO_2_.

The above results are contrary to the total smoke release, where the lowest value was achieved for the PU foam: 761 m^2^/m^2^. The addition of WA particles significantly affects the value of the tested parameter. 10 wt.% of WA particles increase TSR almost twice, i.e., from 761 to 1307 m^2^/m^2^, and almost triple TSR at an amount of 30 wt.%. In turn, the presence of organic matter, which is averagely 4 wt.% and is flammable, and whose share increases with the addition of WA particles, causes an increase in the total amount of smoke released. Such changes in smoke release during the combustion of a material were observed by the authors of [54]. They explained it by the formation of cyclic compounds and unstable carbon particles bonding to residue. Even though the total smoke release is greater for WAPU foam than for PU foam, WA particles do not increase the amount of combustion products in smoke and gas mixture.

As seen from Table 2, 10 wt.% of WA particles does not impact the yields of carbon dioxide and carbon monoxide while the maximum amount of WA particles, i.e., 50 wt.% reduces CO_2_ and CO emission during combustion by 76% and 77% compared to neat PU foam indicating a high suppression capability for toxic gas release. The WA particle barrier first suppresses heat delivery and inflammable pyrolysis products between PU polymer matrix and fire, thus weakening the combustion reaction. Less combust consumption of pyrolysis products indicated lower AVHR, CO_2_, and CO [55].

Additionally, the results LOI are presented in Figure 4. Obviously, PU foam is a flammable material, and its LOI value is only 21.3%. However, some authors [56] present even lower LOI values for unmodified PU foams, i.e., up to 20%. With increasing the amount of WA particles to 10 wt.%, the LOI has no obvious change, whereas for WAPU foams with 30 wt.% and 50 wt.% WA particles, the LOI value shows an increase and reaches to 21.5% and 22.3%, respectively. The obtained results show that irrespective of WA particles amount, the obtained WAPU foams are classified as slow burning materials.

To understand the flame retardant mechanism of PU and WAPU foams, scanning electron microscope imaging after cone calorimeter test was conducted to observe the morphology of ignited outer surface of the samples (Figure 5).

For the char residue of PU foam, it can be observed that the outer surface is quite dense and has spherical carbon residues and few holes, which formed due to the release of combustible gas and low capacity of forming solid char layer during fire (Figure 5a). Although, such residue cannot effectively retard the decomposition of volatiles that feed the flame, note that the PU foam exhibited better char layer-forming capability and AVHR values that the ones modified with SiO_2_ in [57] and tris(1-chloro-propyl)phosphate with modified aramid fibers in [58].

Additionally, 10 wt.% of WA particles does not impact the char residual structure, and it is proved by cone calorimeter test results in Table 4. On the contrary, microstructures of char layers of WAPU-30 and WAPU-50 (Figure 5c,d) are consistent, continuous, and have compact carbon layer, which could have contributed to a barrier properties, assuring inhibition of release and transfer of combustible gases.

The most compact char layer after cone calorimeter test is observed for WAPU-50 foam; it was tested for chemical elemental composition using EDS analysis. The obtained results are also compared to control PU foam (Figure 6).

The residual char of PU foam obtained after cone calorimeter test (Figure 5a), as expected, has no flame retardant elements and only carbon (C), oxygen (O), and nitrogen (N) are detected. Similar observations have been done for pure PU foams by [59] who tested the impact f aluminum phosphate microcapsules on flexible PU foams. However, contrary results can be seen in Figure 5b. The highest amount after fire retardancy test is observed for Si and Ca elements which are assigned for SiO_2_ and CaCO_3_, respectively, in WA particles. These compounds are proved to have a positive effect on flame retardancy of PU foams [18,60]. Therefore, the obtained results confirm that the WA particles act as a flame barrier.

## 4. Conclusions

In this work, PU foam was modified with WA particles, which were then applied as a flame retardant additive. Recycling WA particles in the production of WAPU foams showed that they do not greatly deteriorate the performance properties (thermal conductivity, compressive strength, and water absorption) of polyurethane foams, i.e., the addition of up to 30 wt.% WA particles increases compressive strength by up to 8.3%, decreases average cell size 2-fold, and slightly increases thermal conductivity and water absorption of resultant WAPU foams.

Furthermore, WA particles increase T_5 wt.%_ decomposition temperature by 32 °C and char yield at 600 °C of WAPU foams, indicating better thermal stability properties compared to the PU foam. In addition to all of this, they significantly improve fire resistance properties. It was obtained that due to capability of absorbing heat released, 30 wt.% and 50 wt.% WA particles reduced AVHR by 44% and 69%, respectively. Considerable benefit was determined for carbon dioxide and carbon monoxide yields, which were achieved by less combust consumption of pyrolysis products. Therefore, WA particles reduced CO_2_ and CO yields by up to 76% and 77%, respectively. However, WA particles are not characterized by smoke suppression capability and 50 wt.% of WA particles increase TSR values from 761 m^2^/m^2^ to 1154 m^2^/m^2^. The improvement of fire resistance of WAPU foams was additionally confirmed by LOI test. The flame retardant mechanism was further analyzed through SEM imaging and EDS tests. The results demonstrated that superior flame retardancy of 30 wt.% and 50 wt.% WA particles modified WAPU foams could be attributed to consistent and quality char residues which enables WA particles to serve as a flame barrier.

## Figures and Tables

**Figure 1 polymers-12-00683-f001:**
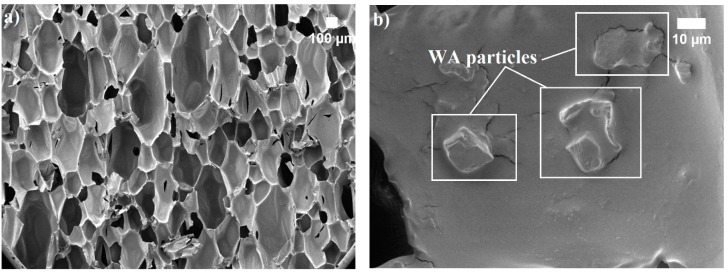
WA particles in WAPU-50 foam structure: (**a**) ×35 magnification; (**b**) ×1000 magnification.

**Figure 2 polymers-12-00683-f002:**
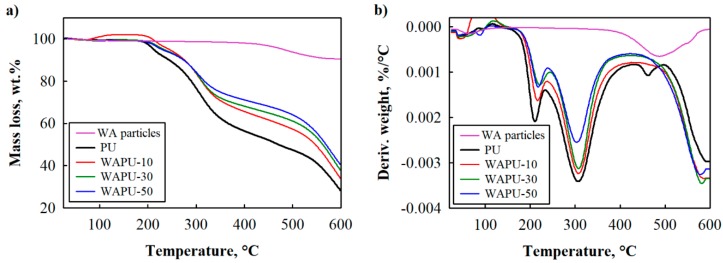
Thermal stability of PU and WAPU foams: (**a**) thermogravimetric analysis (TGA) and (**b**) differential thermogravimetry (DTG).

**Figure 3 polymers-12-00683-f003:**
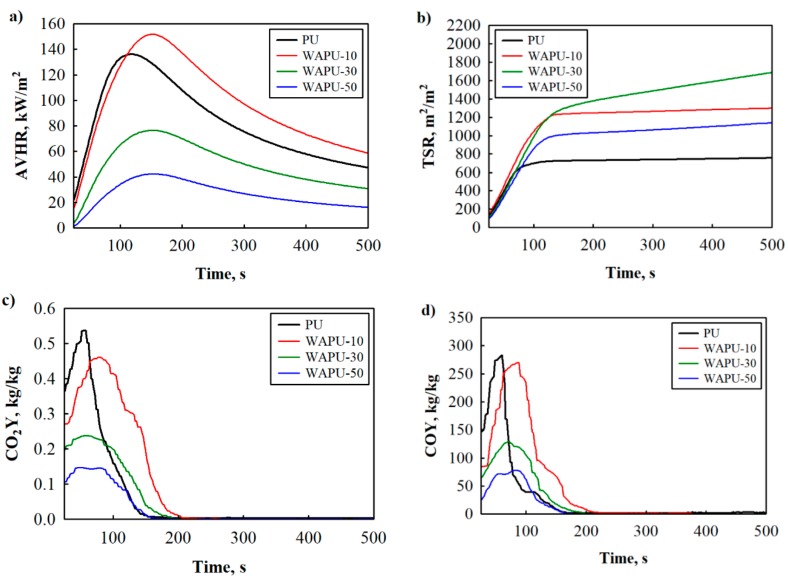
Flame retardancy of PU and WAPU foams: (**a**) average heat emission (AVHR); (**b**) total smoke released (TSR); (**c**) CO2Y and (**d**) COY.

**Figure 4 polymers-12-00683-f004:**
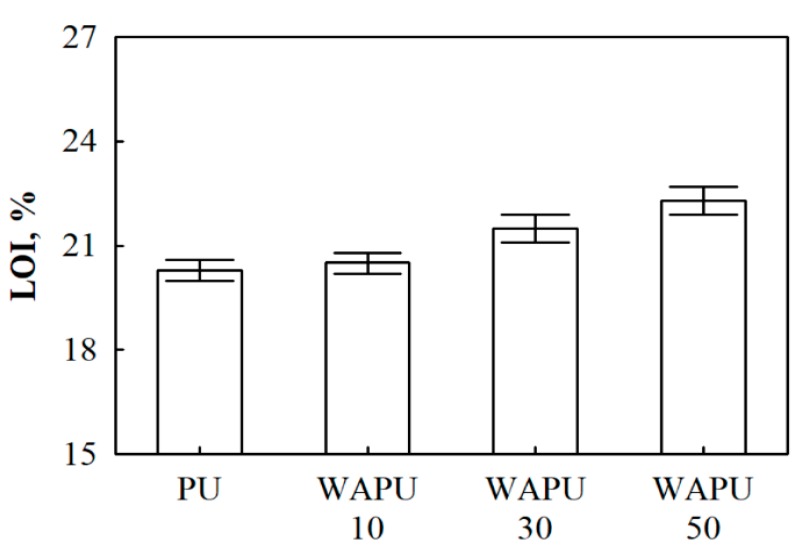
Limited oxygen index (LOI) test for PU and WAPU foams.

**Figure 5 polymers-12-00683-f005:**
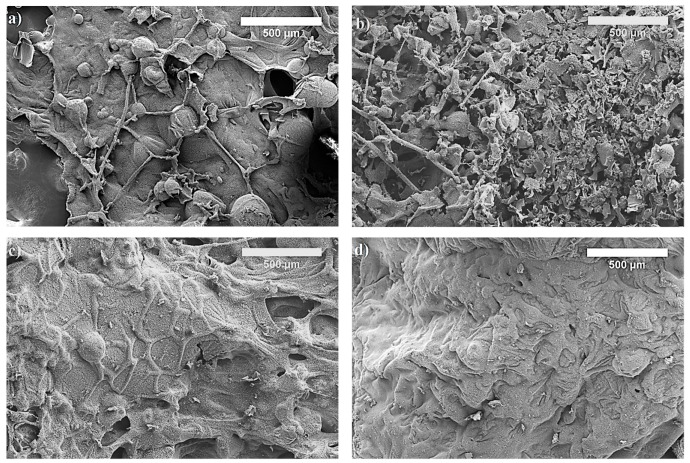
SEM images of PU and WAPU foams char after cone calorimeter tests: (**a**) PU; (**b**) WAPU-10; (**c**) WAPU-30, and (**d**) WAPU-50.

**Figure 6 polymers-12-00683-f006:**
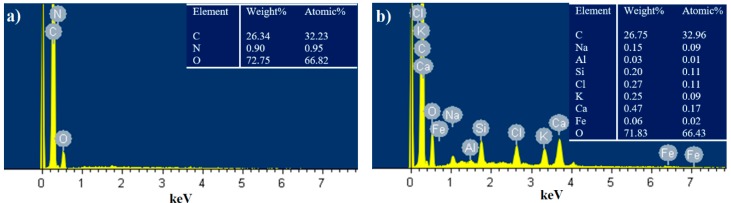
EDS analysis of a char residues of (**a**) PU and (**b**) WAPU-50 foams.

**Table 1 polymers-12-00683-t001:** Formulations of polyurethane (PU) and WA particles modified PU (WAPU) foams.

ComponentComponent	Total Amount, pbw *	
	PU	WAPUU
BioPolyol RDBioPolyol RD	60	60
Petol PZ 400-4G	40	40
Blowing agent	2.7	2.7
Blowing and gelling catalyst	1.0	1.0
Silicone surfactant	2.5	2.5
WA amount, wt.%	0	10 or 30 or 50
Isocyanate index	125	

* Parts by BioPolyol RD and Petol PZ 400-4G total weight.

**Table 2 polymers-12-00683-t002:** Physical and mechanical properties of PU and WAPU foams.

Foam Title	Characteristics				
	Apparent Density, kg/m^3^	Cell Diameter, µm	Thermal Conductivity after 28 days, W/(mK)	Compressive Strength, kPa	Water Absorption, vol.%
PU	38.2 ± 0.4	475 ± 13	0.0336 ± 0.002	240 ± 17	5.2 ± 0.14
WAPU-10	46.5 ± 1.2	218 ± 25	0.0349 ± 0.003	260 ± 29	5.5 ± 0.25
WAPU-30	51.0 ± 0.5	200 ± 30	0.0357 ± 0.002	250 ± 11	5.8 ± 0.55
WAPU-50	56.8 ± 0.6	201 ± 28	0.0361 ± 0.002	233 ± 14	6.5 ± 0.24

**Table 3 polymers-12-00683-t003:** Thermal degradation parameters of PU and WAPU foams from TGA analysis.

Foam Title	T5 wt.%, °C	T50 wt.%, °C	Tmax, °C				Char Yield at 600 °C, wt.%
			1st Stage	2nd Stage	3rd Stage	4th Stage	
PU	213	470	211	309	462	-	28.2
WAPU-10	245	547	217	309	-	590	33.9
WAPU-30	236	562	222	308	-	584	37.8
WAPU-50	230	570	217	301	-	580	40.4

**Table 4 polymers-12-00683-t004:** Flame retardancy parameters of PU and WAPU foams.

Foam Title	Parameter			
	AVHR, kW/m^2^	TSR, m^2^/m^2^	CO_2_Y, kg/kg	COY, kg/kg
PU	136	761	3.92	0.11
WAPU-10	152	1307	4.22	0.14
WAPU-30	76.6	1758	1.69	0.049
WAPU-50	42.5	1154	0.926	0.025

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
