# Peer review of "Fire Suppression and Thermal Behavior of Biobased Rigid Polyurethane Foam Filled with Biomass Incineration Waste Ash"

_polymers, 2020, doi:10.3390/polym12030683_

Round 1

Reviewer 1 Report

Dear authors,
The topic which was choosed in your paper is very meaningful. From my point of view, there are
some results should be explained or added in your manuscript. Here my suggestions:
(1) SEM images of four types of foams should be offered by author, from the SEM, author should
mark the size of typical foam cells. From those SEM images, author may find the relationship of
density with the size of foam cells. in this way, it can also explain the WA contents influence the
density of foams.
(2) It is recommended to test the limiting oxygen index (LOI) of the foam samples.
(3) If author has enough time, the char residue of foams should be tested to obtain the different
contents of the flame retardant elements by EDAX.

Author Response

Authors would like to thank reviewer for his/her efforsts. The comments were very valuable for the improvement of the article. Please find bellow the answers to your remarks:

(1) SEM images of four types of foams should be offered by author, from the SEM, author should mark the size of typical foam cells. From those SEM images, author may find the relationship of density with the size of foam cells. in this way, it can also explain the WA contents influence the density of foams.

Answer: the research of the structure using SEM imaging and calculation of cell size was already done by authors in previous publication in Composites Part A journal. Authors do not want to repeat the results which were already published. However, the dependence of cell size and density of these foams was not observed. 10-50 wt.% WA particles reduce average cells to almost the same size while the extent of apparent density increase with the addition of WA particles is very high.

(2) It is recommended to test the limiting oxygen index (LOI) of the foam samples.

Answer: LOI test was conducted and the results are presented in Figure 4.

(3) If author has enough time, the char residue of foams should be tested to obtain the different contents of the flame retardant elements by EDAX.

Answer: EDS test was conducted and the results are presented in Figure 6.

Reviewer 2 Report

Dear Authors, i find this article quite interesting, below are my remarks and questions.

Line 34 – Grammar

 Together with growing use of biomass-based fuels for energy generation will lead to large amounts of biomass incineration ash which causes serious environmental problems.

Growing use of biomass-based fuels for energy generation will lead to large amounts of biomass incineration ash which causes serious environmental problems.

Line 43 – Grammar

It was shown that WA should be used as a pozzolan in up to 20 wt.% [6–7] and as an additive mainly 43 in 10 wt.% [8] to obtain concrete material with an optimal characteristics as well as reduce CO2

emissions and price [9]. -> It was shown that WA could be used as a pozzolan  up to 20 wt.% [6–7] and as an additive mainly  up to 10 wt.% [8] to obtain concrete material with an optimal characteristics as well as to reduce price and CO2 emissions [9].

 Line 49 – Question

Why water absorption should belong to valuable performance characteristics?

LINE 54 Grammar

and lower price ->and low price

Line 55 Grammar

compounds it emit toxic smoke -> compounds emit toxic smoke.

Line 63 - Style

[21– 22]. [23] observed that the addition of up to 40 wt.% of basalt waste reduces compressive strength of 64 PU composites and worsens their dimensional stability while [24] tested feldspar

Some researchers [21– 22, 23] observed that the addition of up to 40 wt.% of basalt waste reduces compressive strength of 64 PU composites and worsens their dimensional stability while others [24] tested feldspar

Line 66 - Question

However, a study [10] showed that WA particles as waste-based and filler-type additive can be used in higher  amounts, i.e. up to 50 wt.%, because they act as a plasticizer in polyol premix.

Is this correct? How solid, inorganic particles can act as a plasticizer?

Line 86 – Question

Have you characterized the shape and size distribution of the WA particles?

Line 93

Isocianate index -> Isocyanate index

WA amount, wt.%

0

10; 30 and 50

WA amount, wt.%

0

10 or 30 or 50

Line 96 – Logical

Further, 10 wt.%, 30 wt.% and 50 wt.% of WA were added -> Further, 10 wt.% or 30 wt.% or 50 wt.% of WA were added

Line 113 – Grammar – changing present to past tense

For the evaluation of WA particles impact on PU and WAPU foams under high temperatures,  thermogravimetric (TGA) and difference thermogravimetric (DTG) are conducted under the air 114 atmosphere using STA 449 F1 Jupiter Analyzer (Netzsch Group, Germany) at temperature interval  from 25°C to 600°C. Temperature raising speed is 10°C/min. -) For the evaluation of WA particles impact on PU and WAPU foams under high temperatures, 113 thermogravimetric (TGA) and difference thermogravimetric (DTG) analyses  were conducted under the air atmosphere using STA 449 F1 Jupiter Analyzer (Netzsch Group, Germany) at temperature interval from 25°C to 600°C. Temperature raising speed was 10°C/min.

Line 117 – 125: Grammar -  Please change present to past tense and use upper index for square meter

Line 143 - Style

It can be noticed from the thermal conductivity data, WA particles do not significantly increase the parameter of WAPU-50 foam and it is only by 7% higher than that of the PU foam.->

It can be noticed that WA particles do not significantly increase the thermal conductivity of WAPU-50 foam and it is only by 7% higher than that of the PU foam.

Line 144 -

Even through -> Even though

Line 146 – improve this chaotic sentence!

As [38] stated, it can be due enhanced effect of thermal conductivity from the solid parts, in this case WA particles and thicker cell walls, as the cell size decreases under the same blowing  agent contents.

Line 167 Grammar

process consisting of a multiple  physical and chemical reactions -> process consisting of  multiple physical and chemical reactions

Line 177

Additionally, 28.2 wt.% of  char yield at 600°C is obtained.  -> Additionally, 28.2 wt.% of  char at 600°C is obtained.  (or : Char yield at 600°C was 28.2 wt.%)

Line 185 - Question

It  means that WA particles are capable to shift onset degradation temperature towards higher temperature, reduce weight loss and increase thermal stability of WAPU foams. This behaviour can  be attributed to the barrier effect provided by WA particles which reduces both the heat and oxygen fluxes toward the PU foam surface.

Comment: Exactly speaking T5 wt.% is not the onset degradation temperature. The WA particles probably do not influence the thermal degradation of urethane linkages in the hard segments. WA particles are thermally stable and the shift of T5 wt.%, T10 wt.% results mainly lower amount of the polymer in the foam. With the size of samples used for TGA the barrier effect for heat flux is of lower significance, there is no oxidation  at the beginning of thermal degradation of hard and soft segments, the oxidation occurs later at higher temperatures, therefore oxygen diffusion is of no importance for the onset of thermal degradation of PU foams  (how big were the samples? Were the foams crushed to fit into the weighting cell?)  

Have you run the TGA with pure WA for comparison?

Author Response

Authors would like to thank reviewer for his/her efforts. The comments were very valuable for the improvement of the article. Please find bellow the answers to your remarks:

1) Line 34 – Grammar

Together with growing use of biomass-based fuels for energy generation will lead to large amounts of biomass incineration ash which causes serious environmental problems.

Growing use of biomass-based fuels for energy generation will lead to large amounts of biomass incineration ash which causes serious environmental problems.

Answer: corrected.

2) Line 43 – Grammar

It was shown that WA should be used as a pozzolan in up to 20 wt.% [6–7] and as an additive mainly 43 in 10 wt.% [8] to obtain concrete material with an optimal characteristics as well as reduce CO2

emissions and price [9]. -> It was shown that WA could be used as a pozzolan  up to 20 wt.% [6–7] and as an additive mainly  up to 10 wt.% [8] to obtain concrete material with an optimal characteristics as well as to reduce price and CO2 emissions [9].

Answer: corrected.

3) Line 49 – Question

Why water absorption should belong to valuable performance characteristics?

Answer: water absorption is the mandatory property to test for manufacturers in order to get conformity assessment from notified body (sprayed polyurethane foams acc. EN 14315-1 and factory made polyurethane foams acc. EN 13165, Annex ZA).

4) LINE 54 Grammar

and lower price ->and low price

Answer: corrected.

5) Line 55 Grammar

compounds it emit toxic smoke -> compounds emit toxic smoke.

Answer: corrected.

6) Line 63 - Style

[21– 22]. [23] observed that the addition of up to 40 wt.% of basalt waste reduces compressive strength of 64 PU composites and worsens their dimensional stability while [24] tested feldspar

Some researchers [21– 22, 23] observed that the addition of up to 40 wt.% of basalt waste reduces compressive strength of 64 PU composites and worsens their dimensional stability while others [24] tested feldspar

Answer: corrected.

7) Line 66 - Question

However, a study [10] showed that WA particles as waste-based and filler-type additive can be used in higher  amounts, i.e. up to 50 wt.%, because they act as a plasticizer in polyol premix.

Is this correct? How solid, inorganic particles can act as a plasticizer?

Answer: authors would like to apologize, it is a honest mistake. The sentence was corrected.

8) Line 86 – Question

Have you characterized the shape and size distribution of the WA particles?

Answer: the shape and size distribution was characterized in authors previous work published in Composites Part A journal.

9) Line 93

Isocianate index -> Isocyanate index

Answer: corrected.

10)

WA amount, wt.%

0

10; 30 and 50

WA amount, wt.%

0

10 or 30 or 50

Answer: corrected.

11) Line 96 – Logical

Further, 10 wt.%, 30 wt.% and 50 wt.% of WA were added -> Further, 10 wt.% or 30 wt.% or 50 wt.% of WA were added

Answer: corrected.

12) Line 113 – Grammar – changing present to past tense

For the evaluation of WA particles impact on PU and WAPU foams under high temperatures,  thermogravimetric (TGA) and difference thermogravimetric (DTG) are conducted under the air 114 atmosphere using STA 449 F1 Jupiter Analyzer (Netzsch Group, Germany) at temperature interval  from 25°C to 600°C. Temperature raising speed is 10°C/min. -) For the evaluation of WA particles impact on PU and WAPU foams under high temperatures, 113 thermogravimetric (TGA) and difference thermogravimetric (DTG) analyses  were conducted under the air atmosphere using STA 449 F1 Jupiter Analyzer (Netzsch Group, Germany) at temperature interval from 25°C to 600°C. Temperature raising speed was 10°C/min.

Answer: corrected.

13) Line 117 – 125: Grammar -  Please change present to past tense and use upper index for square meter

Answer: corrected.

14) Line 143 - Style

It can be noticed from the thermal conductivity data, WA particles do not significantly increase the parameter of WAPU-50 foam and it is only by 7% higher than that of the PU foam.->

It can be noticed that WA particles do not significantly increase the thermal conductivity of WAPU-50 foam and it is only by 7% higher than that of the PU foam.

Answer: corrected.

15) Line 144 -

Even through -> Even though

Answer: corrected.

16) Line 146 – improve this chaotic sentence!

As [38] stated, it can be due enhanced effect of thermal conductivity from the solid parts, in this case WA particles and thicker cell walls, as the cell size decreases under the same blowing  agent contents.

Answer: improved.

17) Line 167 Grammar

process consisting of a multiple  physical and chemical reactions -> process consisting of  multiple physical and chemical reactions

Answer: corrected.

18) Line 177

Additionally, 28.2 wt.% of  char yield at 600°C is obtained.  -> Additionally, 28.2 wt.% of  char at 600°C is obtained.  (or : Char yield at 600°C was 28.2 wt.%)

Answer: corrected.

19) Line 185 - Question

It  means that WA particles are capable to shift onset degradation temperature towards higher temperature, reduce weight loss and increase thermal stability of WAPU foams. This behaviour can  be attributed to the barrier effect provided by WA particles which reduces both the heat and oxygen fluxes toward the PU foam surface.

Comment: Exactly speaking T5 wt.% is not the onset degradation temperature. The WA particles probably do not influence the thermal degradation of urethane linkages in the hard segments. WA particles are thermally stable and the shift of T5 wt.%, T10 wt.% results mainly lower amount of the polymer in the foam. With the size of samples used for TGA the barrier effect for heat flux is of lower significance, there is no oxidation  at the beginning of thermal degradation of hard and soft segments, the oxidation occurs later at higher temperatures, therefore oxygen diffusion is of no importance for the onset of thermal degradation of PU foams  (how big were the samples? Were the foams crushed to fit into the weighting cell?)  

Have you run the TGA with pure WA for comparison?

Answer: the TGA of pure WA particles was conducted and Figure 2 was supplemented with the obtained results. And yes, the foam was crushed into the weighting cell before the conducting TGA test. Additionally, the amount of polymer in the foam does not reduce with the addition of WA particles. The discussion was a little bit improved.